# A Gold Standard, CRISPR/Cas9-Based Complementation Strategy Reliant on 24 Nucleotide Bookmark Sequences

**DOI:** 10.3390/genes11040458

**Published:** 2020-04-23

**Authors:** François M. Seys, Peter Rowe, Edward L. Bolt, Christopher M. Humphreys, Nigel P. Minton

**Affiliations:** 1Clostridia Research Group, BBSRC/EPSRC Synthetic Biology Research Centre (SBRC), School of Life Sciences, Biodiscovery Institute, University of Nottingham, Nottingham NG7 2RD, UK; francois.seys@nottingham.ac.uk (F.M.S.); pete@deepbranchbio.com (P.R.); c.humphreys@nottingham.ac.uk (C.M.H.); 2School of Life Sciences, Queen’s Medical Centre, Nottingham NG7 2UH, UK; ed.bolt@nottingham.ac.uk; 3NIHR Nottingham Biomedical Research Centre, Nottingham University Hospitals NHS Trust, Nottingham NG7 2RD, UK

**Keywords:** bookmark, CRISPR/Cas9, complementation, *Clostridium*, knock-out

## Abstract

Phenotypic complementation of gene knockouts is an essential step in establishing function. Here, we describe a simple strategy for ‘gold standard’ complementation in which the mutant allele is replaced in situ with a wild type (WT) allele in a procedure that exploits CRISPR/Cas9. The method relies on the prior incorporation of a unique 24 nucleotide (nt) ‘bookmark’ sequence into the mutant allele to act as a guide RNA target during its Cas9-mediated replacement with the WT allele. The bookmark comprises a 23 nt Cas9 target sequence plus an additional nt to ensure the deletion is in-frame. Here, bookmarks are tailored to *Streptococcus pyogenes* CRISPR/Cas9 but could be designed for any CRISPR/Cas system. For proof of concept, nine bookmarks were tested in *Clostridium autoethanogenum*. Complementation efficiencies reached 91%. As complemented strains are indistinguishable from their progenitors, concerns over contamination may be satisfied by the incorporation of ‘watermark’ sequences into the complementing genes.

## 1. Introduction

The genus *Clostridium* encompasses many species of medical and industrial interest, including *Clostridium difficile* [1], *Clostridium botulinum* [2], *Clostridium acetobutylicum* [3], and *Clostridium autoethanogenum* [4]. Accordingly, substantial efforts have been expended in recent years to develop convenient and standardised genetic toolboxes to enable the systematic study and engineering of clostridia [5,6,7,8]. A fundamental requirement has been the ability to establish gene function through precise gene knock-out (KO) and the subsequent phenotypic comparison of the mutant KO and progenitor strain, typically, the wild type (WT). However, before a particular change in phenotype may be definitively assigned to the absence of the disrupted gene, it is essential to carry out a complementation study to demonstrate that the introduction of a functional copy of the deleted gene restores the phenotype of the KO strain to that of the progenitor [9]. This rules out the possibility that other, ancillary mutations are responsible for the observed phenotype rather than inactivation of the targeted gene. However, the complementation method itself can also cause unintended changes in phenotype, particularly if the introduced gene is located on an autonomous, multicopy plasmid. To circumvent this, we have previously reported the use of the *pyrE* locus as a genomic site at which single copies of the gene may be easily inserted [8,10]. By necessity, however, this strategy requires the prior construction of a specific *pyrE* mutant and assumes that the positioning of the gene at this location has no unforeseen effects.

A ‘gold standard’ complementation would ideally involve the restoration of the deleted allele to the WT at its original locus. This may be achieved using ‘bookmark’ complementation—a simple CRISPR/Cas9-based strategy illustrated in Figure 1—relying on homology-directed repair (HDR) [11]. It is reliant on the prior incorporation into the mutant allele of a 24 nt bookmark sequence between the start and stop codon of the gene to be deleted, which represents a single guide RNA (sgRNA) target for the *Streptococcus pyogenes* Cas9 nuclease (SpCas9). The bookmark comprises a 20 nt protospacer directly upstream of a 3 nt protospacer adjacent motif (PAM) and is extended to 24 nt by the addition of a single random nucleotide before the protospacer or after the PAM. As this comprises eight codons, the replacement of the targeted gene by the mutant allele creates an in-frame deletion. The positioning of the single nucleotide either before the protospacer or after the PAM, and/or bookmark orientation relative to the start codon of the KO gene, can be varied to avoid premature stop codons.

When a typical CRISPR/Cas9 vector (incorporating a sgRNA that targets the bookmark) is used to replace the mutant allele with a WT copy of the gene by homologous recombination, the complemented strain generated is immune to the effects of Cas9. In contrast, the genome of the in-frame deletion mutant is cleaved due to the presence of the bookmark. This allows the rapid selection of complemented strains. The complemented strain obtained with the bookmark strategy is theoretically identical to the parental WT strain. As such, a silent mutation, or ‘watermark’, that can be detected by an appropriate polymerase chain reaction (PCR) screen should be inserted into the complementing gene [10]. Its presence eliminates any concerns that the experiment has been contaminated with the WT strain. 

The validation of the principle of bookmark complementation was achieved in three steps. In the first instance, because the bookmark strategy is reliant on the sequence used being an effective target for SpCas9 and being absent in the organism being manipulated, we chose a total of nine different protospacers from three different bacterial species that had been successfully exploited in previous studies as SpCas9 targets. BLASTn (https://blast.ncbi.nlm.nih.gov) was then used to establish that the test organism to be manipulated here, *C. autoethanogenum*, did not contain equivalent sequences, and off-target efficiencies were calculated with the algorithm of Hsu et al. 2013 [12]. In the second step, the *pyrE* gene was replaced by a contiguous array of all nine bookmark protospacers and their respective PAM (*BMa*) using a previously described CRISPR/Cas9 editing system [6,7]. In the third and final step, nine CRISPR/Cas9 vectors incorporating a functional copy of *pyrE* and flanking homology arms, together with one of the nine sgRNAs each targeting a different bookmark protospacer, were used [6,7] to restore the inactivated genomic *pyrE* gene to WT. This particular gene was chosen because of the ease with which the mutant and WT allele can be phenotypically distinguished. The mutant allele confers uracil auxotrophy on the host as well as resistance to 5-fluoroorotic acid (5-FOA). WT strains are in contrast sensitive to 5-FOA and are uracil prototrophs. 

## 2. Materials and Methods

### 2.1. Strains and Medium

*Clostridium autoethanogenum* strain DSM10061 was recovered from cryostocks and cultivated in pre-reduced yeast tryptone fructose (YTF) medium in an anaerobic cabinet (Don Whitley Scientific Ltd, Bingley, UK) at 37 °C [13]. YTF solid medium was then complemented with the following antibiotics: 7.5 µg/mL thiamphenicol for transconjugants selection; 250 µg/mL D-cycloserine for elimination of the *E. coli* DNA donor strain; 1 mg/mL of 5-fluoroorotic acid for *ΔpyrE* mutant selection. A *ΔpyrE C. autoethanogenum* strain was also generated to be used as a control for the colony PCR of the *pyrE* locus. This strain was generated through conjugation with pMTL431511_CLAU_pyrE [7],—a standard pMTL vector [14] with a functional truncated Cas9, sgRNA targeting *pyrE* and editing template which leave only the first two and last two nucleotides of the *pyrE* gene in the *pyrE* locus after successful HDR.

*Escherichia coli* strain K12 DH5α was used as host for vector assembly and cloning and was cultivated in Luria-Bertani (LB) broth with 12.5 µg/mL chloramphenicol for transformant selection. *E. coli* strain sExpress [15] was used as the conjugal DNA donor strain to transfer plasmids into *C. autoethanogenum*. It was grown on LB broth with 50 µg/mL kanamycin to maintain the vector R702, which enables conjugative transfer, with the addition of 12.5 µg/mL chloramphenicol to maintain the vectors to be conjugated into *C. autoethanogenum*.

### 2.2. Basic Local Alignment Search Tool (BLAST) Search

The parameters presented in Table 1 were used to BLAST all nine bookmark protospacers with their respective PAMs (23 bp) against the published *Clostridium autoethanogenum* genome (PubMed ID CP012395.1) [13].

### 2.3. Off-Target Efficiency 

The parameters presented in Table 2 were used to calculate the off-target efficiency of each bookmark protospacer using the Molecular biology suite software suite of Benchling (San Francisco, CA, USA) and the algorithm of Hsu et al. [12].

### 2.4. Vector Assembly

All kits, enzymes, and buffers were purchased from New England Biolabs Ltd (Hitchin, UK) and used following the manufacturer’s instructions unless specified. Primers and single stranded DNA (ssDNA) oligomers were ordered from Sigma-Aldrich Company Ltd (Gillingham, UK) and resuspended in 100 μM solutions with TE buffer (10 mM Tris: 1 mM EDTA; pH 8.0). Unless otherwise specified, all PCR reactions used Q5 high-fidelity polymerase 2x master mix with whole *C. autoethanogenum* cells as DNA template and occurred after a 10 min denaturation at 98 °C, followed by 35 cycles of 10 s denaturation at 98 °C, 15 s annealing at 60 °C, 3 min elongation at 72 °C. Unless otherwise specified, vectors were assembled using HiFi assembly kit. In this case, all parts were designed to share between 25 nt and 40 nt of homologous sequences with their intended adjacent fragments. The sequence of all primers used in this study are available in Appendix A.

The vector pMTL431511_CLAU_pyrE was digested with AsiSI and AscI, then dephosphorylated with Antarctic phosphatase and gel purified to remove its editing template. In parallel, the homology arms of the *pyrE* locus were amplified. The left homology arm was amplified using oFS109 with oFS119 and the right homology arm using oFS112 with oFS120. The amplicons were then run on a 1% (w/v) agarose gel for gel extraction and elution in 6 μL of de-ionised, nuclease free water. The Bookmark array was synthesised by Integrated DNA Technologies (IDT) Ltd (Sheffield, UK) and resuspended in 50 μL de-ionized and nuclease-free H_2_O. Finally, digested pMTL431511_CLAU_pyrE, left homology arm, right homology arm, and bookmark array were assembled together in one single step using a NEBuilder^®^ HiFi DNA assembly cloning kit from New England Biolabs to form the vector pMTL431511_BMa. 

The assembly of the bookmark complementation vectors was done in two steps. First, a new editing template meant to complement *pyrE* was introduced in the vector pMTL431511_CLAU_pyrE. The *pyrE* editing template was amplified using the PCR primers oFS66 and oFS67. After PCR clean-up, the *pyrE* editing template was digested with AsiSI and AscI and gel purified in 6 μL elution buffer. Finally, 50 ng of the digested and dephosporylated pMTL431511-CLAU-pyrE were ligated with 40 ng of the digested *pyrE* editing template at 16 °C overnight using T4 DNA ligase and associated buffer in a 20 μL reaction, to form the intermediary pMTL431511_pyrE_sg3 vector.

In a second assembly step, the intermediary pMTL431511_pyrE_sg3 vector was digested with SalI and dephosphorylated with Antarctic phosphatase and gel purified in 6 μL elution buffer. In parallel, stocks of double-stranded DNA (dsDNA) coding for the sgRNA targeting each bookmark were made by annealing together ssDNA oligomers. ssDNA oligomers annealing was achieved by mixing 5 μL of 100 mM solution of each oligomer (oFS39 to oFS47) with its respective reverse-complementary oligomer (oFS79 to oFS87), then incubating them 5 min at 98 °C, 5 min at 72 °C, and 5 min at 50 °C before holding the temperature at 15 °C. dsDNA sgRNA oligomers were then diluted 100-fold in New England Biolabs buffer 2.1 to form a dsDNA oligomer stock. Finally, the nine bookmark complementation vectors were assembled in parallel by HiFi assembly combining 1 μL of their respective dsDNA bookmark oligomer stock with 50 ng of the pMTL431511_pyrE_sg3 vector previously digested by SalI. These vectors were labelled pMTL431511_BM4 to pMTL431511_BM12 for their associated target bookmark protospacer.

### 2.5. Cloning 

After vector assembly and transformation into 40 μL of chemically competent *E. coli* K12 strain DH5α, single colonies were suspended in 40 μL of sterile deionised water to be stored, used as DNA template for PCR screening, and used as inoculum for an overnight culture. Vectors were extracted from overnight cultures using a plasmid extraction kit and sent to Eurofins Genomics (Wolverhampton, UK) for Sanger sequencing.

### 2.6. Conjugation into C. autoethanogenum 

Conjugative DNA transfer was undertaken following established methods [7,15,16]. Briefly, 1 mL of a *E. coli* sExpress strain carrying the target plasmid was grown to an OD600 of 0.4, centrifuged at 3000 G for 3 min, washed with 500 µL phosphate-buffered saline solution (PBS), transferred in an anaerobic cabinet, and mixed with 0.2 mL of a *C. autoethanogenum* culture grown overnight to an OD600 of approximately 0.2. The cell mixture was spread on solid YTF without antibiotic and incubated for 24 h. Afterward, cells were resuspended with 600 µL PBS and spread onto YTF plates with 250 µg/mL D-cycloserine and 7.5 µg/mL thiamphenicol, respectively for counter-selection of the sExpress strain and selection of the *C. autoethanogenum* transconjugant. Single colonies were then screened by colony PCR.

### 2.7. Colony Polymerase Chain Reaction (PCR) 

After conjugation of a vector into *C. autoethanogenum*, eight individual colonies were patched onto YTF selective medium to isolate mutants. After four days, the patches that grew were screened by PCR using the primers oFS105 and oFS106 and Q5 polymerase from New England Biolabs, along with a WT and a *ΔpyrE* strain of *C. autoethanogenum* as positive controls. Amplicons of the expected size were then purified using a QIAquick PCR Purification Kit from Qiagen (Manchester, UK) and validated by Sanger sequencing by Eurofins Genomics.

### 2.8. Plasmid Loss and Cryopreservation

Once a mutant strain had been confirmed by PCR, it was re-streaked on YTF plates without antibiotic to isolate single colonies. Single colonies were then re-streaked again, this time both on YTF medium with and without thiamphenicol selection. Colonies that failed to grow under thiamphenicol selection were further confirmed to have lost the vector by colony PCR with oFS68 and oFS75. Finally, the strain was inoculated into liquid YTF media and incubated for three days and 0.9 mL of culture used to generate cryostocks with 0.1 mL of dimethyl sulfoxide. These were then stored at –80 °C.

## 3. Results

### 3.1. Bookmark Design

Nine protospacer sequences were taken from successful examples of SpCas9 mutagenesis [17,18,19] and assigned an arbitrary PAM. Using BLAST to look for similar sequences in the genome of *C. autoethanogenum* (accession CP012395.1) highlighted no sequences identical to any of the 23 nt bookmark candidates, Appendix A. The most similar hits were 19 non-consecutive identical bases out of 23 in BM9 (in position 223466) and 18 non-consecutive identical bases in BM11 (in position 3324583) and BM5 (in positions 1235296, 2165840 and 4340901). The off-target score of each bookmark protospacer was >98%, which confirmed them as suitable bookmark protospacers (Appendix A). An extra nucleotide was then added to either extremity of each bookmark candidate to carry their length to 24 nt and avoid the generation of an internal stop codon. The final bookmark sequences are summarised in Table 3.

### 3.2. pyrE Knock-Out and Bookmarks Knock-In

The native *pyrE* gene of *C. autoethanogenum* was replaced with a mutant allele comprising the contiguous 207 nt array of nine protospacer sequences and their respective PAM (Appendix A), flanked by the first and last two codons of the *pyrE* gene. This was done using pMTL431511_BMa, a KO vector based on a previously described CRISPR/Cas9 vector which expresses a truncated Cas9 nuclease (trCas9) [7]. The editing template of pMTL431511_BMa comprises a left homology arm (LHA) and a right homology arm (RHA) that flank the protospacer array. The LHA essentially covers the 1 kb of DNA from the region upstream of *pyrE* but includes the first two codons of the gene. The RHA represents the 1 kb region downstream of *pyrE*, beginning with the last two codons of the gene. The sgRNA targets a region only present in the WT *pyrE* allele.

After conjugation of pMTL431511_BMa into *C. autoethanogenum* (Figure 2A, Appendix A), eight individual colonies were patched onto 5-FOA selective medium to isolate Δ*pyrE*::*BMa* mutants. After four days, six patches had grown and were screened by colony PCR using the primers oFS105 and oFS106. Cells derived from WT and Δ*pyrE* strains of *C. autoethanogenum* were used as controls. Four colonies generated an amplified DNA fragment close to the predicted size of 1.7 kb (Figure 2B), albeit weakly in the case of lanes 3 and 4. Sanger sequencing of the amplified DNA fragment of a randomly selected clone (lane 1) confirmed the replacement of *pyrE* with the bookmark array. The chosen strain was re-streaked on plates without antibiotic to lose the KO vector before storage in cryostocks.

### 3.3. Bookmark Complementation of pyrE

Having generated a KO strain carrying a *pyrE* mutant allele, it was restored to WT using CRISPR/trCas9 vectors (pMTL431511_BM4 to pMTL431511_BM12) which harboured one of nine sgRNA cassettes targeting each of the bookmark sequences (Appendix A) and a 2.5 kb region encompassing the entire *pyrE* gene as editing template (Figure 2C). Following transfer of these plasmids into the Δ*pyrE*::*BMa* strain, colony PCR was undertaken on the transconjugants obtained to ascertain whether they carried the WT or mutant allele (Appendix A). To calculate the complementation efficiency of each bookmark protospacer, the number of colonies that reverted to WT after conjugation was divided by the total number of colonies screened (Appendix A). Each complementation was independently performed three times (Figure 2D). 

The average complementation efficiency was 91 ± 15%, with BM9 having consistently the maximal efficiency (out of seven, seven, and five colonies screened for each replicate respectively). The lowest efficiency was achieved by BM12, with one of its three replicates achieving only 33% efficiency (only two out of six colonies were successfully complemented); however, the other two replicates of BM12 achieved 100% efficiency (four out of four colonies were complemented in both cases). Taken together, these results show that all bookmark complementation vectors are probably equivalent in their ability to target the protospacers used in each genomic bookmark and to replace them by a gene such as *pyrE*.

## 4. Discussion

The combined data obtained established the proof of principle for a new gold standard complementation strategy that is broadly applicable to any organism compatible with CRISPR/Cas9-mediated homology-directed mutagenesis. By complementing the gene of interest back into its original locus, bookmark complementation avoids the biases of traditional complementation methods such as an extra metabolic burden or the disruption of an additional genomic locus. Additionally, the bookmarks are designed to produce in-frame mutations without internal stop codons so as to minimise the chances of inducing polar effects. Here, we describe and characterise a small library of nine functional protospacers that were highly effective as bookmarks for complementation in *C. autoethanogenum*. However, they would be equally applicable to any organism which lack homologous sequences in their genome. 

It is clear that bookmark technology will prove useful beyond complementation, such as during the sequential assembly of large genomic operons or to assist various genome editing strategies [20].

## Figures and Tables

**Figure 1 genes-11-00458-f001:**
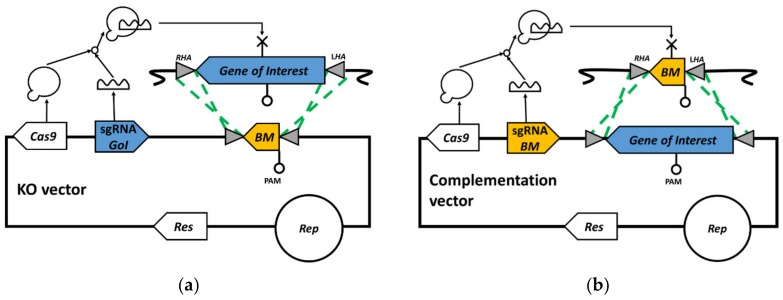
Overview of the bookmark complementation strategy. (**a**) First step—knockout of the gene of interest and insertion of the bookmark in its genomic locus using a knock-out vector consisting of a Cas9 nuclease and single guide RNA (sgRNA) expression cassettes, as well as an editing template composed of one 24 nucleotides (nt) bookmark flanked by homology arms. (**b**) Second step—after isolation of the knockout (KO) mutant and plasmid loss, another round of Cas9-mediated homology-directed mutagenesis is carried out with the help of a complementation vector, to restore the gene of interest in its original locus. The complementation vector is identical to the KO vector, except for its sgRNA cassette, which targets the genomic bookmark that was previously inserted, and editing template, which consists of the gene of interest flanked by the same homology arms. The gene of interest can be watermarked with a silent mutation for higher reliability of the complementation step. GoI: Gene of interest, Res: Antibiotic resistance marker, Rep: Replicon, BM: Bookmark, LHA: Left homology arm, RHA: Right homology arm.

**Figure 2 genes-11-00458-f002:**
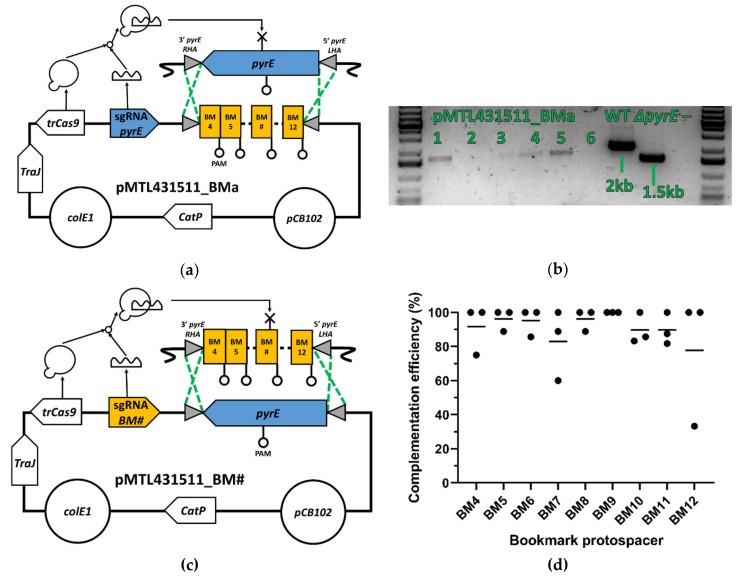
Proof of principle of the bookmark strategy. (**a**) Integration of an array of bookmark protospacers into the genomic locus of *pyrE* using pMTL431511_BMa. The bookmark protospacer array (BM4 to BM12) is flanked by two homology arms of 1 kb, each homologous to the genomic region directly upstream (LHA) and downstream (RHA) of the *pyrE* locus and which include the first two and last two codons of *pyrE*. Homology directed mutagenesis of the *pyrE* locus replaces the *pyrE* gene with the protospacer array. trCas9 nuclease and a sgRNA expression cassettes counter-select wild type (WT) transconjugants by cutting the genomic DNA of cells which have a WT *pyrE* locus. (**b**) Electrophoresis gel of six *C. autoethanogenum* colonies obtained after conjugation of pMTL431511_BMa. The *pyrE* locus of each colony was amplified using the primers oFS105 and oFS106 then run on a 1% (w/v) agarose gel. The expected size of the amplicon of a successfully knocked-in bookmark array (BMa) is 1.7 kb, versus 2 kb for the WT *pyrE* locus and 1.5 kb for a clean Δ*pyrE* genotype without the bookmark array. No DNA template was added to the PCR mix of the negative control (–). (**c**) Integration of the *pyrE* gene back in its original locus using pMTL431511_BM# to target different bookmark protospacers. Each homology arm consists of a 1 kb region directly upstream (LHA) or downstream (RHA) of the *pyrE* locus, including the first two and the last two nucleotides of the *pyrE* gene, respectively. Together with the *pyrE* gene they constitute the *pyrE* editing template. trCas9 nuclease and sgRNA expression counter-selects *pyrE*Δ::*BMa* transconjugants by cutting the genomic DNA of cells with an intact bookmark protospacer array. (**d**) Complementation efficiency of nine protospacers in *C. autoethanogenum*. Complementation of a *pyrE*Δ::*BMa* strain of *C. autoethanogenum* using different bookmark protospacers was successful in 91 ± 15% of all screened colonies, with little to no variations observed in between the particular heterologous protospacers targeted by each bookmark complementation vectors. (•): complementation efficiency of one separate conjugation, (–): arithmetic mean of the complementation efficiency of three independent conjugations for each bookmark, *TraJ*: conjugative transfer function, *colE1*: Gram-negative replicon, *CatP*: chloramphenicol/thiamphenicol resistance cassette, *pCB102*: Gram-positive repliconBM#: Protospacer sequence of any bookmark, where # is an integer between four and 12.

**Table 1 genes-11-00458-t001:** Parameters of the basic local alignment search tool (BLAST) of all the bookmark protospacers and their associated protospacer adjacent motif (PAM) against *Clostridium autoethanogenum* genomes.

Search Parameter	Value
Program	BLASTn
Word size	7
Expect value	1000
Hitlist size	100
Match/Mismatch scores	1, –1
Gapcosts	5, 2
Filter string	F
Genetic Code	1
**Database**	
Posted date	Mar 24, 2020 9:05 AM
Number of letters	8,771,469
Number of sequences	18
Entrez query	Includes: *Clostridium autoethanogenum* DSM 10061 (taxid:1341692)

**Table 2 genes-11-00458-t002:** Parameters used to calculate the off-target efficiency (specificity score) of all nine bookmarks using the Molecular biology suite software suite of Benchling and the algorithm of Hsu et al. [12].

Parameter	Value
Reference genome	GCA_000484505.1 (*Clostridium autoethanogenum* DSM 10061)
Design type	Single guide
Guide length	20 bp
PAM	NGG (SpCas9, 3’side)

**Table 3 genes-11-00458-t003:** Summary of the bookmark sequences. Nine protospacers were picked from the literature as successful examples of SpCas9 targets in bacterial species [17,18,19]. To constitute a bookmark, each protospacer should be immediately followed by a PAM (suggested: AGG) and complemented with one single nucleotide at either extremity. For each bookmark, the position of the extra nucleotide as well as the orientation of the bookmark relative to the coding sequence (CDS) it replaces are given to avoid internal STOP codons. “+”: same direction as the target CDS, “ – ”: reverse-complementary direction relative to the target CDS, “+/−”: either same or reverse-complementary direction relative to the target CDS, *S. pneumonia: Streptococcus pneumonia, B. subtilis: Bacillus subtilis, L. reuteri* = *Lactobacillus reuteri.*

Bookmark	Bookmark Sequence (24 nt)				Orientation	Origin
	Extra nt	Protospacer (20 nt)	PAM	Extra nt		
BM4	G	AGGGTTGTGGGTTGTACGGA	AGG	/	+/−	*S. pneumonia*
BM5	/	ATTTCTGATATTACTGTCAC	AGG	A	+/−	*S. pneumoniae*
BM6	/	ACCGATACCGTTTACGAAAT	AGG	A	+/−	*S. pneumoniae*
BM7	G	TGAAGATCAGGCTATCACTG	AGG	/	+	*B. subtilis*
BM8	G	TCCGGAGCTCCGATAAAAAA	TGG	/	+/−	*B. subtilis*
BM9	G	TATTGATTCTCTTCAAGTAG	AGG	/	−	*B. subtilis*
BM10	/	CCATTGTACTATCATGCTAG	AGG	A	+/−	*L. reuteri*
BM11	G	ATGCAGTCGGCTGTAGAAAG	AGG	/	+/−	*L. reuteri*
BM12	G	CGACTGCATTTTATTATGTA	AGG	/	+/−	*L. reuteri*

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
