# Peer review of "A Gold Standard, CRISPR/Cas9-Based Complementation Strategy Reliant on 24 Nucleotide Bookmark Sequences"

_genes, 2020, doi:10.3390/genes11040458_

Round 1

Reviewer 1 Report

It is nice work describing a new complementation strategy that is applicable to organisms "compatible with CRISPR/Cas9-248 mediated homology-directed mutagenesis", as the authors state. The manuscript is clearly written and edited, the figures are clear, the description of the results and the discussion go to the point. To method extends the genetic tool box that the group has developed over the years.

I have not major criticisms. Perhaps the authors could comment on two points:

1) the size of the homology arm, for example in the case of the pyrE locus, is of 1kb, both for the LHA and RHA pieces. What would be the minimum size that would still function efficiently? 

2) would the method work for genes that are part of operons? and what the strategy be inn that case?

Reviewer 2 Report

In this study, Francois M Seys and colleagues proposed a CRISPR/Cas9-based complementation approach in C. autoethanogenum using 24 nucleotide bookmark sequences as efficient and versatile technology of genome editing. This strategy is reliant on the prior incorporation into the mutant allele of a 24 nt bookmark sequence between the start and stop codon of the gene to be deleted (in this case pyre locus), which represents a sgRNA target for the Streptococcus pyogenes Cas9 (SpCas9) nuclease. The bookmark comprises a 20 nt protospacer directly upstream of a 3 nt Protospacer Adjacent Motif (PAM) and is extended to 24 nt by the addition of a single random nucleotide before the protospacer or after the PAM, in order to obtain an in-frame deletion. When a typical CRISPR/Cas9 vector (incorporating a sgRNA that targets the bookmark) is used to replace the mutant allele with a WT copy of the gene by homologous recombination, the genome of the in-frame deletion mutant is cleaved due to the presence of the bookmark. This bookmark strategy is also reliant on the fact that the employed sequence is not present in the manipulated organism and it is an effective target for SpCas9. Accordingly, the Authors chose three different protospacers from three different bacterial species that had been successfully exploited as SpCas9 targets.

The functionality of each protospacer in C. autoethanogenum and the validation of the principle of bookmark complementation was achieved in two steps. In the first instance, the pyrE gene was replaced by a contiguous array of all nine bookmark protospacers and their respective PAM (BMa). In the second step, nine CRISPR/Cas9 vectors incorporating a functional copy of pyre and flanking homology arms, together with one of the nine sgRNAs each targeting a different bookmark protospacer, were used to restore the inactivated genomic pyrE gene to WT. This particular gene was chosen because of the mutant and WT alleles can be easily phenotypically distinguished. The mutant allele confers uracil auxotrophy on the host as well as resistance to 5-Fluoroorotiv acid (5-FOA). WT strains are in contrast sensitive to 5-FOA and are uracil prototrophs.

The strength of this study as it has been proposed to the Editor of Genes as Concept paper is the detailed description of the strategy employed and the material and methods adopted to achieve the reported results. However, the manuscript lack the full description of the experiments performed to fulfil all the aims reported in the Introductions.

Therefore, I will list below the critical points that should be accomplished in order to make the manuscript acceptable for publication:

  1. In the introduction the Authors stated that BLAST was used to establish that autoethanogenum do not contain in its genome the final bookmark sequences reported in the manuscript, but they did not report the criteria used to run the BLAST analysis. What is the off-targets scoring for these bookmark sequences?
  2. In the first section of the Results, the Authors states that after conjugation of pMTL431511_BMa into autoethanogenum, 8 individual colonies were patched onto 5-FOA selective medium to isolate DpyrE::BMa mutants and after 4 days, 6 patches had grown and were screened by colony PCR. It is stated that 4 colonies generated and amplified DNA fragment of the predicted size of 1.7kb. However, from the electrophoresis assay reported in Figure 2B the fragment of the predicted size is visible only in the lines 1 and 4. Is there any chance to screen more colonies? Could the Authors explain the reason why they did not sequence all the amplified DNA fragments of the clones instead to sequencing only one clone?
  3. In the second section of the Results, the Author stated that they performed the colony PCR on the transconjugants obtained to ascertain whether they carried the WT or mutant allele. Could the Authors show these additional colony PCR results in order to proof the success of the transconjugation?
  4. Moreover, as reported in the table S4 (supplementary data), is there any reason that could explain the difference between the colony counts of each bookmark protospacer and how could it affect the complementation efficiency.

Reviewer 3 Report

This manuscript showed the complementation of gene knock-outs with studying the complementation efficiencies of several types of 24 nt ‘bookmark’ sequence (as a guide RNA target during its Cas9-mediated replacement with the WT allele). However, the major concern is the necessity to perform such a research. Actually, the CRISPR-Cas9 system generates clean deletion of the particular gene. As the author mentioned, the exact complementation of the mutant will be actually the wild type strain (by inserting a single copy of gene back into the original locus in the genome). Therefore, by comparing the mutant (with clean gene deletion) vs. the wild type, we can draw the exact conclusion we want. Thus, there is no need to do such a complementation in this sense. The author did mention the ‘ancillary mutations’. However, this would not likely occur in the genome engineering process using CRISPR-Cas9. Because the CRISPR-Cas9-based genome engineering process is basically a selection of Cas9 again the wild type unedited cells for the positive mutant. If there is off target(s) for the CRISPR-Cas9, it will simply kill the cells and won’t likely generate the ‘ancillary mutations’. On the other hand, even ‘ancillary mutations’ could be generated during the CRISPR-Cas9-based genome engineering process, then how do you ensure there is no such ‘ancillary mutations’ generated during your CRISPR-Cas9-based complementation process?

In addition, here are some other comments:

  1. Line 184-192 in Page 5, it seems that the 207 nt array of nine protospacer sequences with the first and last two codons can form a small protein. Have you measured the effect of this small protein?
  2. Figure 2b in Page 6, the marker should be marked using ‘-kb’ (and also in the supplementary figure S1 and S2). It is unclear to find out the difference between lane 1-6 and pyrE deletion mutant. Why don’t you label the figure as you did for Figure S2?
  3. Figure 2d in Page 6, it's hard to understand this figure, you can just use one data for each bookmark protospacer with error bars to be clear. By referring the data from Table S4, the screened numbers is not stable and too little for confirming the complementation efficiency, Thus the data showed big difference even for the same BM in Figure 2d. It is better to screen more colonies in order to avoid this issue.
  4. The strain names should be italic in the references.

Round 2

Reviewer 2 Report

The authors have made considerable effort to give adequate response to my concerns I had  and they significantly improved the revised submission.

Reviewer 3 Report

The authors addressed all my comments, and now the manuscript can be published.